# Sex Impact on Knee and Ankle Muscle Extensor Forces during Loaded Running

**Kade D. Wagers** , **Nicholas J. Lobb, AuraLea C. Fain, Kayla D. Seymore and Tyler N. Brown** *

Department of Kinesiology, Boise State University, Boise, ID 83702, USA
* Correspondence: tynbrown@boisestate.edu; Tel.: +1-208-426-5613; Fax: +1-208-426-1894

**Abstract:** Background: This study determined whether the knee and ankle muscle extensor forces increase when running with a body-borne load and whether these forces differ between the sexes. Methods: Thirty-six (twenty male and sixteen female) adults had the knee and ankle extensor force quantified when running 4.0 m/s with four body-borne loads (20, 25, 30, and 35 kg). Peak normalized (BW) and unnormalized (N) extensor muscle force, relative effort, and joint angle and angular velocity at peak muscle force for both the ankle and the knee were submitted to a mixed model ANOVA. Results: Significant load by sex interactions for knee unnormalized extensor force ($p = 0.025$) and relative effort ($p = 0.040$) were observed, as males exhibited greater knee muscle force and effort than females and increased their muscle force and effort with additional load. Males also exhibited greater ankle normalized and unnormalized extensor force ($p = 0.004$, $p < 0.001$) and knee unnormalized force than females ($p = 0.005$). The load increased the normalized ankle and knee muscle force ($p < 0.001$, $p = 0.030$) and relative effort ($p < 0.001$, $p = 0.044$) and the unnormalized knee muscle force ($p = 0.009$). Conclusion: Running with a load requires greater knee and ankle extensor force, but males exhibited greater increases in muscle force, particularly at the knee, than females.

**Keywords:** load carriage; musculoskeletal injury; muscle force; sex dimorphism; biomechanics





## 1. Introduction

Running with a body-borne load is a common service member training and operational activity. During these activities, service members routinely run with heavy body-borne loads (often greater than 20 kg) that reportedly alter lower limb biomechanics, increasing the incidence of musculoskeletal injury [1,2]. In fact, approximately 75% of service members sustain an overuse musculoskeletal injury during training or other occupational activities [3]. Most military-related overuse injuries are sustained in the lower limb, with over 80% occurring at the knee or ankle [4]. Treating the bone and/or soft-tissue damage of a knee or ankle musculoskeletal injury is expensive, yet often fails to prevent re-injury and medical discharge for the service member [5,6]. Considering knee and ankle musculoskeletal injury (and likely re-injury) may pose the greatest threat to military readiness, particularly for female service members, who are twice as likely to suffer a musculoskeletal injury than their male counterparts [7], it is imperative to understand the explicit changes in lower limb biomechanics that increase injury risk when running with heavy body-borne load [8].

When running with heavy military body-borne load, individuals reportedly alter the lower limb biomechanics, which may elevate injury risk of the passive knee and ankle structures. To run with loads greater than 20 kg, individuals produce larger, faster vertical ground reaction forces (vGRFs) that require increases in knee and ankle torsional stiffness up to 19% and 6% [9,10]. While these large increases in joint stiffness may be necessary to maintain stability and prevent limb collapse when running with heavy load [11], it may further increase injury risk [12]. A stiffer joint, for instance, may limit flexion, leading to decreased energy absorption by the musculature, allowing greater transmission of vGRFs to the musculoskeletal system and elevating injury risk [13,14]. Considering

individuals typically increase lower limb flexion moments when running with load, the larger vGRFs may also stem from greater force contributions from the knee and ankle support musculature [15,16].

During locomotion, the knee musculature must generate sufficient force to prevent limb collapse, while the ankle musculature must produce force to propel the center of mass forward [16,17]. However, these muscle force demands may depend on the speed, length, and mode of locomotion [18]. Individuals produce greater knee and ankle extensor muscle force with each incremental increase in walk speed (from −5% to +5% of normal), and the ankle extensors operate at greater relative effort (typically near 100% of maximal capacity) than the knee extensors (between 63% and 72%, respectively) at faster locomotor speeds [19–21]. Considering running with body-borne load increases knee and ankle muscular activity and peak joint moments, it may also require greater knee and ankle muscle force to maintain limb stability and forward propulsion [22,23]. Substantial increases in knee and ankle muscle force, however, may increase the likelihood of lower limb musculoskeletal injury by placing "extra" stress onto the musculoskeletal system [24]. Although running with heavy body-borne load requires greater knee and ankle muscle activity and joint moments, it is currently unknown whether individuals exhibit similar increases in muscle force.

Female service members are typically smaller and weaker than their male counterparts, which may lead to larger changes in the lower limb biomechanics and elevated injury risk when running with heavy body-borne load [25,26]. Brown et al., for instance, recently reported females adopted a 15% stiffer knee and used approximately 5% less peak knee flexion than the male participants to run with heavy loads [9]. Considering that with each 1 Nm/deg increase in knee stiffness, recreational runners are 18% more likely to suffer musculoskeletal injury, and an extended knee may increase the likelihood of soft-tissue injury, females' knee biomechanics may contribute to their high rate of injury [27,28]. However, it is unclear if females exhibit dimorphism in knee and ankle muscle force to run with heavy, military body-borne loads. Therefore, the purpose of this study was to: (1) determine whether relative knee and ankle muscle force increase when running with body-borne load and (2) whether a sex dimorphism in knee and ankle muscle force exists for running with load. We hypothesized that knee, but not ankle, relative effort would significantly increase with each incremental addition of a body-borne load, and females would exhibit greater knee and ankle relative muscle force than males.

## 2. Materials and Methods

### 2.1. Participants

We recruited 36 (16 female and 20 male) adults between 18 and 40 years old to participate (Table 1). Each potential participant was physically active (determined as ≥560 on the physical activity readiness questionnaire) and self-reported the ability to carry up to 75 pounds (~34 kg) [29], but participants were excluded if they reported: (1) a neurological disorder; (2) a history of previous back or lower extremity surgery; (3) pain in the back or lower extremity prior to testing; (4) and/or a recent back or lower extremity injury (previous six months). Research approval was obtained from the local IRB, and written informed consent was provided to participate.

**Table 1.** Mean (SD) age, height, and weight for the male and female participants.

| | N | Age (Years) | Height (m) [a] | Weight (kg) [a] |
|---|---|---|---|---|
| Males | 20 | 21.5 (2.8) | 1.8 (0.1) | 82.6 (11.6) |
| Females | 16 | 21.2 (2.8) | 1.7 (0.1) | 65.0 (11.5) |

[a] Denotes a significant main effect of sex.

### 2.2. Load Configurations

Each participant completed an over-ground running task with four different body-borne loads (20, 25, 30, or 35 kg). For all body-borne loads, participants wore spandex top

and shorts, weighted vest (Box, WeightVest.com, Rexburg, ID, USA), and standard issue military helmet (ACH), as well as carried a mock military weapon (M16) (Figure 1). Using 1 kg weights, the vest weight was systematically adjusted to apply the load necessary for each condition. The vest was weighed prior to testing to ensure only loads +/− 2% of the target were applied. To minimize the effects of fatigue, testing with each load was separated by at least 24 h. To randomize and counter balance testing, the load testing order was assigned to each participant using a 4 × 4 Latin Square design prior to data collection.

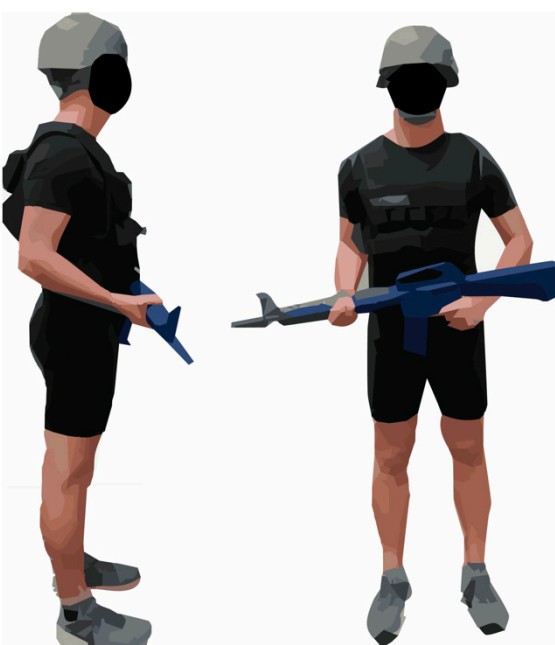

**Figure 1.** Depicts the equipment configuration for each load condition (20 kg, 25 kg, 30 kg, or 35 kg). For each condition, participants were outfitted with a helmet, mock weapon, and weighted vest, which was systematically adjusted to provide the load necessary for each condition.

### 2.3. Over-Ground Walk Task

During data collection, participants had three-dimensional (3D) lower limb joint biomechanics data recorded during an over-ground running task. The running task required participants to run approximately 10 m at 4 m/s ± 5% through the motion capture volume. For each run, a single force platform (2400 Hz, OR6, AMTI, Watertown, MA, USA) and eight high-speed optical cameras (240 Hz, MXF20, Vicon, Oxford, UK) recorded biomechanics data. During each run, two sets of infrared timing gates (TF100, TracTronix, Lenexa, KS, USA) were used to quantify the running speed. Participants completed three successful trials of the running task. A successful trial required participants to run at the correct speed and only contact the force platform with their dominant limb, which was determined as the foot they prefer to kick a ball with. To minimize the fatigue during testing, participants were given water and provided adequate rest between each trial.

### 2.4. Biomechanical Analysis

For each successful run trial, lower limb biomechanics were quantified from the 3D coordinates of 34 retro-reflective markers using Visual 3D (v6.00, C-Motion, Rockville, MD, USA), in accordance with previous work [30]. For processing, synchronous 3D GRF and marker trajectories were low-pass filtered using a fourth order Butterworth filter (12 Hz). Then, Visual 3D processed the filtered marker trajectories to solve sagittal plane lower limb joint rotations using a joint coordinate system approach [31]. To obtain external sagittal plane lower limb joint moments, the filtered kinematic and GRF data were submitted to standard inverse dynamics analyses, with inertial properties established by Dempster [32]. The external joint moments were normalized to subject mass (kg) and

height (m). All biomechanical data were normalized from 0% to 100% of the stance phase, which was defined as heel strike to toe-off (first instance GRF exceeded and fell below 10 N, respectively).

A custom-written MATLAB (Mathworks, Natick, MA) code calculated the muscle force for the ankle and knee extensors, according to a previous work [33,34]. For both the ankle and the knee, sagittal plane joint angle ($x_{ankle}$ and $x_{knee}$) and net unnormalized joint moments ($M_{ankle}$ and $M_{knee}$) were submitted to the muscle force analysis. First, at each joint, the effective extensor moment arm ($L_{ankle}$ and $L_{knee}$) was calculated as a function of the joint flexion angle using the nonlinear equation (Equations (1) and (3)), and then, the extensor muscle force ($F_{ankle}$ and $F_{knee}$) was calculated by dividing the net joint moment by the effective extensor moment arm (Equations (2) and (4)).

$$L_{ankle} = 2.606E^{-4}\,x_{ankle}^{2} + 0.08297\,x_{ankle} - 0.5910 \tag{1}$$

$$F_{ankle} = M_{ankle}/L_{ankle} \tag{2}$$

$$L_{knee} = 8.0\,E^{-5}\,x_{knee}^{3} - 0.013\,x_{knee}^{2} + 0.28\,x_{knee} + 0.046 \tag{3}$$

$$F_{knee} = M_{knee}/L_{knee} \tag{4}$$

The ankle and knee muscle forces were calculated in Newtons as well as normalized to participant BW. The relative effort of the ankle and the knee extensor musculature was also determined by normalizing the peak ankle and knee extensor muscle force exhibited with each load (20 kg, 25 kg, 30 kg, and 35 kg) to the peak muscle force produced at the respective joint with the 35 kg load condition.

### 2.5. Statistical Analysis

The dependent variables submitted for analysis included normalized (BW) and unnormalized (N) extensor muscle force, relative effort, and sagittal plane joint angle and angular velocity at peak muscle force for both the ankle and the knee. Each dependent variable was averaged across three successful run trials to create a participant-based mean and then submitted to a mixed model analysis of variance to test the main effects and interaction between the body-borne load (20, 25, 30, and 35 kg) and sex (female and male). Significant interactions were submitted to a simple effects analysis, and a Bonferroni correction was used for multiple comparisons. Independent t-tests were used to compare the demographics between the sexes. All statistical analysis was performed using the SPSS v25 software (IBM, Amonk, NY, USA), with an alpha level of 0.05.

### 3. Results

The males were taller ($p < 0.001$) and heavier ($p < 0.001$) but not older ($p = 0.690$) than females (Table 1).

The ANOVA revealed significant load by sex interactions for unnormalized extensor force ($p = 0.025$) and relative effort ($p = 0.040$) at the knee (Table 2 and Figures 2 and 3). The males increased the knee extensor force and relative effort with the 25 and 30 compared to the 20 kg load ($p < 0.023$, $p < 0.028$), while the females exhibited no significant difference in the extensor force or relative effort between any loads ($p > 0.05$). Compared to females, males exhibited greater knee extensor force with the 25, 30, and 35 kg loads ($p < 0.048$) and knee relative effort with the 30 kg load ($p = 0.030$).

The males exhibited greater ankle normalized and unnormalized extensor force ($p = 0.004$, $p < 0.001$) but only greater knee unnormalized force than the females ($p = 0.005$) (Table 2 and Figure 2). At the peak muscle force, the males exhibited less knee flexion ($p = 0.002$) but greater knee flexion angular velocity ($p = 0.001$) than the females, while the females exhibited greater ankle dorsiflexion than the males ($p = 0.018$) (Table 3). Sex had no impact on the ankle or knee relative effort ($p > 0.05$).

**Table 2.** Mean (SD) ankle and knee extensor force for male and female participants with each body-borne load (20, 25, 30, and 35 kg).

| | | 20 kg | | 25 kg | | 30 kg | | 35 kg | |
|---|---|---|---|---|---|---|---|---|---|
| | | **Male** | **Female** | **Male** | **Female** | **Male** | **Female** | **Male** | **Female** |
| Extensor Force (N) | Ankle [b] | 2018.5 (570.1) | 1410.6 (327.2) | 1999.6 (387.0) | 1429.0 (320.5) | 2033.9 (427.7) | 1454.3 (322.9) | 2134.2 (454.8) | 1522.6 (291.0) |
| | Knee [a,b,c] | 3346.6 (843.8) | 2842.7 (625.4) | 3736.2 (900.6) | 2880.1 (820.8) | 3921.2 (888.0) | 2912.5 (731.0) | 3606.0 (843.4) | 3068.4 (691.8) |
| Extensor Force (BW) | Ankle [b,c] | 5.46 (0.85) | 4.77 (0.69) | 5.49 (0.64) | 4.84 (0.64) | 5.59 (0.69) | 4.90 (0.82) | 5.88 (0.77) | 5.12 (0.81) |
| | Knee [c] | 10.66 (1.88) | 10.50 (1.16) | 11.61 (1.91) | 10.53 (1.59) | 11.94 (1.39) | 10.65 (1.69) | 11.18 (1.72) | 10.86 (1.30) |

[a] Denotes a significant load by sex interaction. [b] Denotes a significant main effect of sex. [c] Denotes a significant main effect of load.

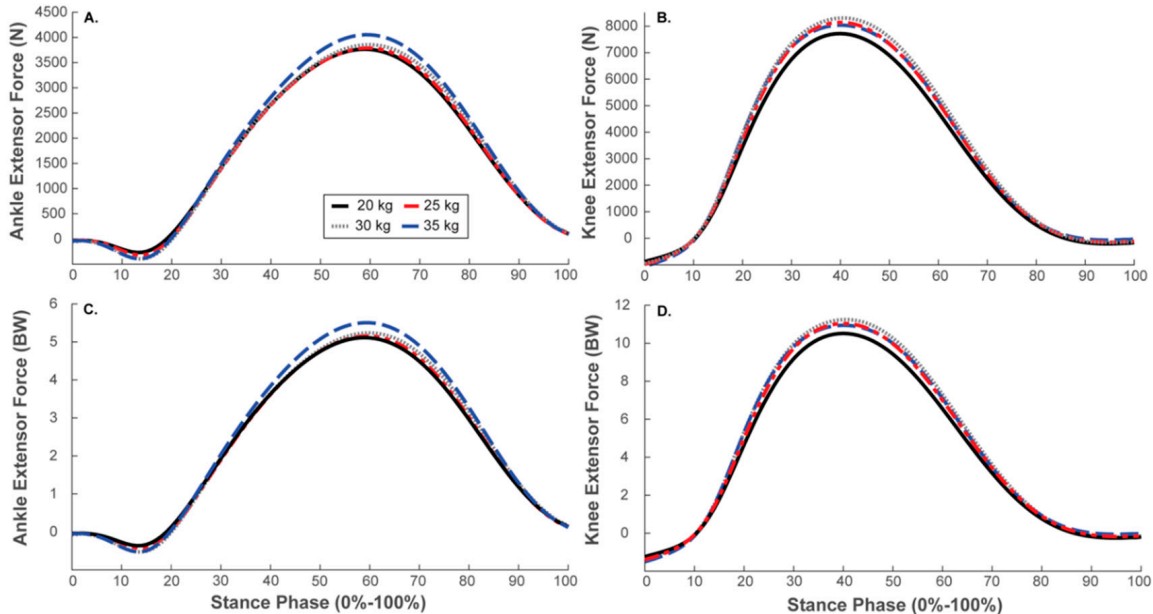

**Figure 2.** Mean stance phase (0–100%) ankle (**A**,**C**) and knee extensor muscle force (**B**,**D**) with each body-borne load (20 kg, 25 kg, 30 kg, and 35 kg).

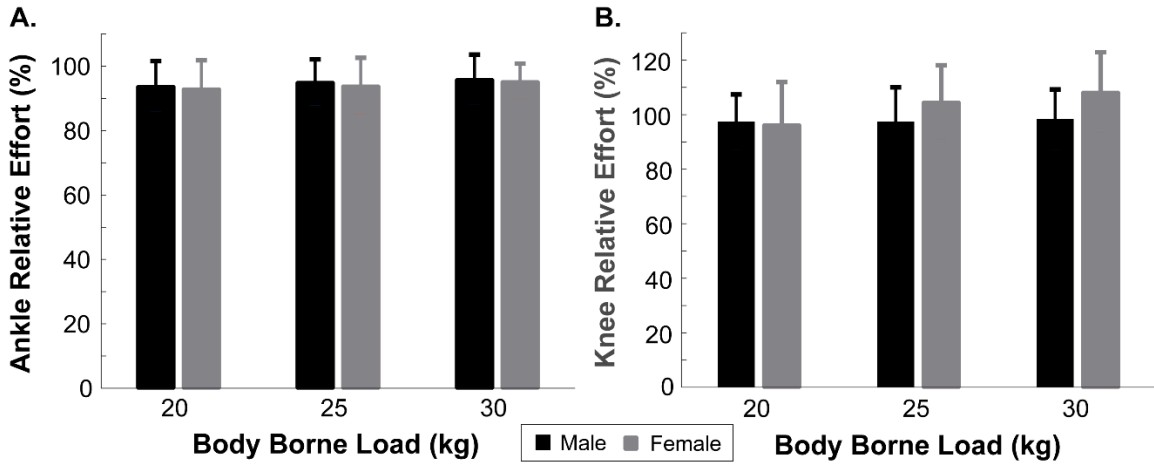

**Figure 3.** Mean ± SD ankle (**A**) and knee (**B**) relative effort (%) for males and females with each body-borne load (20 kg, 25 kg, 30 kg, and 35 kg).

**Table 3.** Mean (SD) ankle and knee sagittal plane joint angle and angular velocity at peak muscle force for male and female participants with each body-borne load (20, 25, 30, and 35 kg).

| | | 20 kg | | 25 kg | | 30 kg | | 35 kg | |
|---|---|---|---|---|---|---|---|---|---|
| | | **Male** | **Female** | **Male** | **Female** | **Male** | **Female** | **Male** | **Female** |
| Joint Angle (deg) | Ankle [a] | 18.97 (7.62) | 24.36 (6.50) | 17.92 (7.39) | 22.65 (8.80) | 18.79 (7.86) | 24.96 (6.98) | 17.92 (6.95) | 24.99 (7.83) |
| | Knee [a] | 42.15 (5.52) | 48.83 (5.95) | 42.65 (5.65) | 48.27 (5.68) | 43.38 (5.38) | 48.24 (5.12) | 41.32 (5.76) | 47.64 (4.77) |
| Angular Velocity (rad/s) | Ankle | −23.24 (34.4) | −17.33 (37.3) | −26.72 (47.0) | −6.69 (21.62) | −21.17 (56.7) | −10.34 (25.1) | −14.86 (38.3) | −6.81 (36.0) |
| | Knee [a] | −52.93 (32.0) | −26.29 (38.6) | −67.50 (37.9) | −26.67 (45.9) | −68.79 (46.6) | −27.09 (45.6) | −62.34 (49.1) | −17.19 (27.3) |

[a] Denotes a significant main effect of sex.

Body-borne load impacted the normalized ankle and knee muscle force ($p < 0.001$, $p = 0.030$) and relative effort ($p < 0.001$, $p = 0.044$) and unnormalized knee muscle force ($p = 0.009$) (Table 2 and Figures 2 and 3). Normalized ankle extensor force and relative effort were greater with the 35 compared to the 20, 25, and 30 kg loads ($p < 0.002$, $p < 0.002$). Both the normalized and unnormalized knee forces were greater with the 30 compared to the 20 kg load ($p = 0.048$, $p = 0.036$), but after correcting for type I error, there was no difference in the knee relative effort between any loads ($p > 0.05$). The load did not impact the ankle unnormalized muscle force or the ankle/knee angle ($p > 0.05$).

## 4. Discussion

Running with a heavy military load led to large increases in the knee and ankle muscle force. Participants used greater knee and ankle extensor force (up to 7.8% and 6.5%, respectively) to run when a body-borne load increased from 20 to 35 kg. These large knee and ankle extensor forces reportedly increase the stress on the lower limb's soft tissue structures and may elevate their risk of sprain, strain, and/or rupture [24]. Specifically, an elevated extensor force may increase the tensile stress on the associated tendons or result in joint biomechanics that produce postures and/or torques that elevate stresses on the joint's ligament structures, leading to greater likelihood of soft-tissue injury. For example, large increases in knee extensor force purportedly increase the patellofemoral joint stress and anterior cruciate ligament loads [28,35]. Running with a body-borne load reportedly requires large increases in the knee and ankle joint moments in conjunction with substantial decreases in joint flexion [9]. The larger joint moments may coincide with the greater knee and ankle extensor force currently observed to prevent limb collapse and a reduction in flexion motion that reportedly limits the active energy absorption by each joint's musculature [13,36]. Consequently, greater extensor force may be produced to safely run with a load, but a larger percentage of these elevated forces may be transferred to the musculoskeletal system, further stressing the soft-tissue structures and elevating injury risk.

In agreement with our hypothesis, running with a load shifted the relative effort proximally to the knee extensors. Typically, during unloaded running, the relative effort of the knee and ankle extensors is 63% and 84% of the maximum, and the ankle extensors operate at a greater relative effort than the knee extensors, regardless of the locomotor speed [20,21]. Yet, contrary to unloaded locomotion, the current participants exhibited a relative knee extensor effort near or greater than the maximum (97% to 103%, respectively), which was between 4% and 7% above the ankle's effort (currently between 93% and 96% of the maximum). While the reason for the current discrepancy with unloaded running is not immediately evident, encumbering individuals with heavy body-borne loads may require a substantial increase in the knee effort, as little muscular reserve is available from the ankle extensors. For instance, older adults, who typically exhibit weaker lower limb musculature—particularly at the ankle—exhibit similar large, albeit not significant, 18%

increase in the relative effort from the knee musculature to run compared to their younger counterparts [37]. The proximal knee musculature operating at or near the maximal effort may be a consequence of lower limb biomechanical alterations that individuals make to run with a heavy load. As mentioned above, during loaded running, individuals purportedly adopt a stiffer knee, from reduced knee flexion in conjunction with large increases in the knee flexion moment, but they also take shorter, more frequent strides with larger, faster vGRFs [10,38,39]. Although the extended knee may increase the mechanical advantage of the associated musculature, it may still require greater knee extensor effort (i.e., force) to provide the adequate muscular support during weight acceptance and prevent limb collapse from the additional body-borne load [18]. In addition, the shorter strides may be a compensatory action when running with a load, which is necessary to afford the individual the ability to reduce the knee flexion and increase the mechanical advantage of the associated musculature. However, this results in a significant increase in strides to complete a military exercise, which allows more impact forces to be transmitted to the musculoskeletal system, contributing to a greater cumulative stress being placed on the joint's soft tissues and increasing the likelihood of service member musculoskeletal injury [40,41].

Contrary to our hypothesis, the males exhibited greater lower limb muscle force when running with a load than females. The males, in fact, produced 25% and 40% more knee and ankle extensor force than the females. However, only the ankle muscle force was greater for males when accounting for body weight. Considering the currently tested males were, on average, 17.6 kg heavier and 0.1 m taller than the females, and larger individuals typically have greater muscle mass, the current males may possess the physical ability (i.e., muscle strength and mass) to increase the knee and ankle extensor force to run with a load [25]. The current males were, indeed, stronger, producing 29.0% (or 1.57 Nm per kg of BM) greater maximum knee extensor torque than the females (see Supplementary Material). This knee extensor strength may afford the males the ability to produce greater muscle force than the smaller females to run with a load and provide them the physical ability to increase the knee extensor force and relative effort up to ~17% and ~12% with the addition of heavy body-borne loads. Stronger service members are reportedly less likely to suffer an operational-related musculoskeletal injury. This physical ability may afford the individual lower limb biomechanics, such as the greater knee flexion velocity currently exhibited by the males (reportedly helping in active energy absorption), which may promote economical energy storage and force generation by the knee extensors [42] and protect the musculoskeletal system during operational activities [15,43,44]. In addition, the males' greater ankle extensor force may increase their ability to produce work for forward propulsion, improving their load carriage performance [45]. Conversely, the weaker females exhibited greater ankle dorsiflexion (purportedly constraining active energy absorption) when running with a load. The females' limited musculature may require them to adopt biomechanics that aid the ankle extensors with work production necessary for forward propulsion [16,45], but this reduces their running economy and exposes their musculoskeletal to greater injury risk during operational activities [15,46]. Future work, however, is needed to determine whether body size and strength, rather than sex, contribute to the dimorphism in lower limb biomechanics and subsequent injury rates between male and female service members. Larger, strong service members may possess the muscular "reserve" to safely attenuate the impact forces placed on the musculoskeletal system and optimize the musculo-tendon dynamics during occupational activities.

A potential limitation is only analyzing the knee and ankle muscle force, as the hip musculature may play an important role in an individual's ability to safely run with a heavy body-borne load [47]. The chosen body-borne loads may limit the study. Not including an unloaded condition may obfuscate the comparison with previous research [20,21], while different equipment configurations, such as a rucksack or different mock weapon, may alter the load distribution and subsequent lower limb biomechanics. Finally, the chosen participants may be a limitation. Although participants self-reported the ability to safely carry body-borne loads, they were not required to have load carriage experience and

thus may have altered their muscle forces compared to the experienced load carriers. Additionally, each participant was physically active, but specific cardiorespiratory and/or muscular fitness levels were not assessed and may have impacted the lower limb when running with a load. Considering the service members' fitness level may predispose them to higher injury rates, its impact on lower limb biomechanics during military load carriage warrants further exploration.

## 5. Conclusions

In conclusion, running with a body-borne load may increase service member injury risk, as it requires large increases in the knee and ankle extensor force to safely complete the activity. Specifically, to run with a heavy body-borne load, participants exhibited an approximate 7% increase in the knee and ankle force and operated both the knee and the ankle near the maximum relative effort. Yet, contrary to unloaded running, the knee operated at greater relative effort than the ankle extensors. The larger, stronger males produced 25% and 40% more knee and ankle extensor force than the females and may possess the physical ability or muscular strength to increase the knee extensor force with the addition of a load.

**Supplementary Materials:** The following supporting information can be downloaded at: https://www.mdpi.com/article/10.3390/biomechanics2030032/s1, Table S1: Mean (SD) dominant knee flexion and extension strength measures (Nm/kg). References [48,49] are cited in the supplementary materials.

**Author Contributions:** Conceptualization, T.N.B.; methodology, K.D.W., N.J.L., A.C.F., K.D.S. and T.N.B.; validation, K.D.W., N.J.L., A.C.F., K.D.S. and T.N.B.; formal analysis, K.D.W., N.J.L., A.C.F. and T.N.B.; investigation, K.D.W., N.J.L., A.C.F. and K.D.S., resources, T.N.B.; data curation, N.J.L., A.C.F., K.D.S. and T.N.B.; writing—original draft preparation, K.D.W. and T.N.B.; writing—review and editing, K.D.W., N.J.L., A.C.F., K.D.S. and T.N.B.; visualization, K.D.W. and T.N.B.; supervision, T.N.B.; project administration, T.N.B.; funding acquisition, T.N.B. All authors have read and agreed to the published version of the manuscript.

**Funding:** This research was supported by Battelle Energy Alliance/Idaho National Laboratory and Natick Soldier Research Development and Engineering Center.

**Institutional Review Board Statement:** The study was conducted according to the guidelines of the Declaration of Helsinki and approved by the Institutional Review Board of Boise State University (protocol number: 103-MED15-008 and date of approval: 10/16/2015).

**Informed Consent Statement:** Informed consent was obtained from all participants involved in the study.

**Data Availability Statement:** The data presented in this study are openly available in the following data repository [doi: 10.18122/cobr_data/3/boisestate].

**Conflicts of Interest:** None of the authors demonstrate any conflicts of interest regarding this submission.

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
