# Peer review of "Sex Impact on Knee and Ankle Muscle Extensor Forces during Loaded Running"

_2673-7078, doi:10.3390/biomechanics2030032_

Round 1

Reviewer 1 Report

General remarks

This is an interesting study especially for the military milieu. The methodology is robust except for the use of absolute loads as compared to relative loads since the study's focus is on sex-related differences. I suspect during military service training both male and female trainees are subjected to the same load carriage, however in my opinion this is a limitation of the study.

Another aspect that could be improved is the use of shorter sentences with clear messages without repetitions, please see detailed comments below.

General comment about introduction

It is far too long and the authors' way of repeating a part of the previous sentence to introduce the following sentence makes it feel that there are many unnecessary repetitions. The study's first hypotheses about an increase in the knee and ankle extensor muscle force with body-borne load is unnecessary. The authors have presented previous evidence supporting this hypothesis, so unless they rephrase it, it is suggested that is removed. The focus of the study is the possible sex-related differences in the knee and ankle muscle force when running overground with a significant body-borne load and its implications for the higher injury rates in female soldiers. It is also recommended that the references used in support of this hypothesis are updated, since they could strengthen their arguments.

p.1, line 31: correct to "...increasing the incidence..."

p.1, lines 39-42: it is suggested that the references associated with the higher incidence of injuries in female army members be at the end of that secondary sentence rather at the end of the sentence at line 43.

p.2, lines 44-62: the message of this paragraph is not clear to me. I also found a couple of repetitions when going from one sentence to the next, which are not necessary. The authors have also used the reference by Williams DS et al., 2004 which is in my opinion unrelated to their story, since that study focused on runners with high- and low-arched feet. 

p.2, lines 44-45: this sentence is a repetition so it is suggested to be removed and the paragraph starts with a shorter introduction to the ensuing biomechanical information.

Methods

p.3, line106: was the physical fitness level of participants assessed in detail? previous studies have reported that the women's lower fitness level could be a factor predisposing them to higher injury rates.

p.4, line 135: correct to "...data recorded during an over-ground...".

p.4, line 135: correct to "...limb joint kinematic data...". Biomechanics refers to a combination of usually recorded kinematic and kinetic data. Do the same in line 148.

p.4, line 135: it is recommended that "run task" be replaced with "running task" everywhere in the manuscript.

p.4, line 136: correct to "...required partcipants to run...".

p.5, line 187: correct to "...main effects and interaction...". 

Discussion

p.10, lines 240-243 and lines 244-246: they formulate the same argument so it is suggested that one of the two is removed.

p.10, 2nd paragraph of discussion: the evidence and arguments presented here are very interesting. I suggested that the 1st hypothesis of the study is replaced by one refering to possible changes in the relative effort of the lower limb musculature as a function of increasing the body-borne load. The results from the knee and ankle extensors force could be then used to support the authors' arguments about how the lower limb musculoskeletal biomechanics adjust to the challenging conditions of heavy load during a running task in the military environment.

p.10-11, 3rd paragraph: what is the point of presenting force in absolute values? Since, the load in the vest was not added as a percentage of individual body weight, any comparisons other than normalized to body weight make no sense. The finding of the greater normalized ankle muscle force for males is interesting and I would expect the authors to comment on that given the different joint angle velocity profiles between men and women here. From Table 3 it can be seen that men relied mostly on the knee musculature to complete the running tasks compared to women whose difference between knee and ankle angular velocity in each load was smaller.

The "Berlin" group by Dr.Arampatzis has produced a lot of nice work on the force generation and efficiency of work production during running. For example, 

on their paper "Muscle-specific economy of force generation and efficiency of work production during human running", Elife 2021, they provide evidence about the distinct roles of vastus lateralis and soleus muscles during running. 

The authors are suggested to consult the group's work in order to get ideas other than the higher physical fitness level of men compared to women that could account for the differences in their results.

p.11, last paragraph of discussion: 

I agree with The limitations presented here, except for the first one. Another serious limitation is not using load normalized to body mass and the authors are invited to comment on that.

Also, since the knee extensor and flexor muscle strength was assessed, why didn't the same happen with the ankle plantarflexors and dorsiflexors?

Table 3, lines 231-233: please add the measure unit for joint angle.

Reviewer 2 Report

Sex Impact on Knee and Ankle Muscle Extensor Forces during Loaded Running

Overall the paper is well written and has a clear focus, there are a couple of places where a more detailed description should be included. 

Figure 1 - it will be good to include a discussion about the weight differences, is it due to different types of helmets/vests, or in some cases, some gears are not included. It is logical to think that some gear (e.g. helmet) affects the ability to move disproportionally to others (e.g. vest). 

Figure 2 - many things need to be explained and controlled. What are the criteria for choosing people: are they the same BMI? same heigh? same weight? same ethics? What variables are controlled when comparing sex and what variables were not? How many people are recruited? 

Figure 3 - what methods did you decide to evaluate knee relative effort? Also, a lot more statistical details need to be included: What variables are controlled when comparing sex, and what variables were not? How many people are recruited? 

Round 2

Reviewer 1 Report

Dear authors,

the revised manuscript has been significantly improved.  Very good  work done. The authors have carefully addressed all my comments. Specifically with regards to the limitation of absolute body-borne load both for male and female service members, I understand your argument about choosing absolute and not relative body-borne loads for the running task and since the audience will probably be associated -but not necessarily- with the military environment, it is desirable to proceed like that. 

In my opinion, the manuscript is now eligible to be considered for publication in the journal.